

**Seismic anisotropy under Zagros foreland from SKS splitting observations**
**Khalil Motaghi[1], Ayoub Kaviani[2], Wathiq Abdulnaby[3], Hanan Mahdi[4], Haydar Al-Shukri[4]**
[1]Department of Earth Sciences, Institute for Advanced Studies in Basic Sciences (IASBS), Zanjan 45137-66731,
Iran
[2]Institute of Geosciences, Goethe University, Frankfurt, Germany
[3]Seismological Laboratory of University of Basrah (SLUB), Department of Geology, College of Science,
University of Basrah, Basrah, Iraq
[4]University of Arkansas at Little Rock (UALR), Little Rock, Arkansas, US
**Correspondence:** Khalil Motaghi (khalil1024@yahoo.com)
**Abstract**
We present SK[K]S splitting measurements from 18 newly deployed seismic stations in the
foreland of the Zagros collision zone, providing new insights into asthenospheric flow and
lithospheric deformation associated with the Arabian-Eurasian continental collision. Our
results reveal two distinct fast-axis orientations: NE-SW in northern Iraq and NW-SE in the
Mesopotamian Plain and Persian Gulf. The NE-SW anisotropy in northern Iraq aligns with
fast-axis orientations observed in the Iranian-Anatolian Plateau and the azimuth of absolute
plate motion, indicating large-scale asthenospheric flow as the primary influence across the
northern Middle East. In contrast, the NW-SE anisotropy in the Mesopotamian Plain and
Persian Gulf, characterized by smaller splitting times, parallels previously reported Pn
anisotropy, suggesting a contribution from lithospheric mantle anisotropy, likely a remnant of
past rifting. The influence of asthenospheric flow on the observed seismic anisotropy in this
region appears minor. These findings refine our understanding of mantle dynamics and
lithosphere-asthenosphere interactions in the Zagros collision zone.
**Keywords:** Anisotropy, Zagros collision, Mesopotamia, Asthenospheric flow, SK[K]S



## 1. Introduction

Seismic anisotropy is an important tool for investigating the dynamics of the Earth's mantle and the processes driving plate tectonics (e.g., Park and Levin, 2002; Long and Becker, 2010). It provides insights into deformation within the crust and upper mantle, and helps detect large-scale deep features such as mantle flows that are not easily identified by other geophysical methods (Silver and Chan, 1991). In this study, we focus on the foreland region of the Zagros continental collision zone, aiming to develop a geodynamic model that elucidates the interaction between mantle flow and continental lithosphere. Specifically, we investigate mantle anisotropy to characterize the interplay between the globally northeast-directed asthenospheric flow and the lithospheric root beneath the Zagros orogeny (Priestley et al., 2012). The dynamics of the lithospheric mantle in this region are shaped by shear and normal tractions associated with large-scale plate motions and localized mantle processes (Sandvol et al., 2003). This research represents the first investigation of mantle anisotropy within the foreland basin of the Arabia-Eurasia plate boundary, encompassing eastern Iraq and the Persian Gulf (Fig. 1).

Seismic anisotropy in the mantle lithosphere, asthenosphere, or both, can yield teleseismic shear-wave splitting (Silver and Chan, 1991; Silver and Holt, 2002). In the Middle East, anisotropy studies based on splitting analysis of core-refracted phases (SKS and SKKS) have revealed a consistent NE-oriented asthenospheric flow beneath the Anatolian plate and northern Iran, corresponding to the global mantle flow in the no-net rotation (NNR) reference frame (Kaviani et al., 2021; Paul et al., 2014; Arvin et al., 2021). Near the southwestern edge of the Arabian plate, this flow is deflected northward by the Afar plume (Hansen et al., 2006). However, the Zagros collision zone in western Iran exhibits a more intricate anisotropy pattern,





influenced by deformation of the thick Arabian lithosphere (Priestley et al., 2012; Sadegh-
Bagherabadi et al., 2018b) and deflected asthenospheric flow in response to significant
variations in lithospheric thickness compared to adjacent regions (Kaviani et al., 2021). Despite
the significance of anisotropic features in this area, the Zagros foreland remains underexplored
due to logistical challenges. The partial coverage of the region by the Persian Gulf complicates
seismic data acquisition (Fig. 1), and eastern Iraq has historically lacked adequate seismic
monitoring. To address this knowledge gap, this study uses new seismic data from a regional
network of 17 stations in Iraq and a newly established station in the Persian Gulf. The Iraqi
stations, established as part of a network enhancement initiative by the Lawrence Livermore
National Laboratory (LLNL) of the U.S. Department of Energy, provide observations from a
previously under-sampled region. Using shear wave splitting analysis of SKS and SKKS
phases, we investigate the interaction between the asthenospheric flow beneath the Arabian
plate and the thick lithospheric root at its northern edge, focusing on the Zagros foreland region.
This study addresses two primary scenarios for interaction between the asthenospheric flow
and the lithosphere root beneath the Zagros foreland. First, the thick lithospheric root may force
the asthenospheric channel to greater depths, allowing the asthenospheric flow to continue
northwestward in alignment with the NNR frame motion. Alternatively, the lithosphere root
may act as a barrier, redirecting the asthenospheric flow toward shallower regions beneath
adjacent thinner lithosphere, such as northwest Iran, the Anatolian plate, or the oceanic plate
of the Makran subduction zone. In addition to asthenospheric flow, we also consider the
possibility of "frozen" anisotropy, inherited from the study region's tectonic history as part of
the northern Gondwana landmass.

**2. Data and method**





A new regional seismic network was installed in Iraq (Fig. 1) through a collaboration with the
University of Arkansas at Little Rock (UALR). Data acquisition began in 2014 through a multi-
institutional collaboration (Abdulnaby et al., 2020). Table 1 provides the station coordinates
and the time intervals during which data were accessible. Most stations were repositioned over
time, with data collection periods ranging from five months to nine years (Table 1). To enhance
lateral coverage in the Zagros foreland, we also incorporated data from a newly established
seismic station situated on Khark Island in the Persian Gulf. This station is operated by the
International Institute of Earthquake Engineering and Seismology in Iran.
We extracted three-component waveforms from teleseismic earthquakes with magnitudes $\geq 6.0$
from epicentral distances between 90° and 140°. A total of 3256 records met these criteria.
Two splitting parameters—Φ (fast-axis anisotropy orientation) and δt (splitting time between
fast and slow polarizations)—were estimated using the rotation-correlation method of Bowman
and Ando (1987). Before performing the splitting analysis, we visually examined the
waveforms to confirm low noise levels and to ensure that the SK[K]S phases were not distorted
by other teleseismic phases with similar arrival times. Band-pass filtering was applied using
visually selected cutoff periods, with low cutoff periods ranging from 5 to 10 s and high cutoff
periods ranging from 20 to 30 s.
We manually selected the analysis window around the theoretical SK[K]S onset, calculated
using the IASP91 standard velocity model (Kennett and Engdahl, 1991). The window length
was chosen to include at least one period of the clearly observable target phases. Final splitting
parameters were retained in the dataset after meeting the following quality criteria: (1) a signal-
to-noise ratio > 2 for the radial component within the analysis window, (2) a minimum
correlation coefficient > 0.90 between the fast and slow components, (3) elliptical particle
motion before correcting for anisotropy and nearly linear particle motion after correction, (4)
a measured splitting time exceeding 0.5 s, and (5) at least a 50% reduction in energy on the





transverse component after anisotropy correction using the calculated splitting parameters, in
cases of non-null measurements. Measurements showing initial linear particle motion were
classified as null, indicating no splitting.
Figure 2 presents two examples of the splitting analysis, illustrating the energy on the original
transverse (Sh) component and the elliptical particle motion of the horizontal components, both
indicative of shear wave splitting. Main criteria for reliable measurements include the
observation of linear particle motion and significant energy reduction on the Sh component
following correction using the estimated splitting parameters. The final dataset includes 155
reliable non-null measurements and 630 null measurements, as shown in Figures 3 and 4.

**3. Results**
Figure 3 shows the rose diagrams of both null and non-null measurements for 18 new stations
in the Zagros foreland. Figure 4 presents the rose diagrams for non-null splitting measurements
at seismic station locations, with individual measurements (represented by red bars) projected
to their piercing points, alongside measurements from previous studies at a depth of 200 km.
At stations AMR1, AMR2, BSR2, NSR1, NSR3, NSR4, ANB1, and KIR1, the non-null
measurements exhibit a consistent unimodal pattern (Fig. 3). To explore the origin of the
bimodal fast-axis orientations, individual non-null measurements projected to their piercing
points provide valuable insights (Figure 4b). The results reveal distinct fast-axis orientations
east and west of station SLY1, as well as north and south of station DHK1, both situated near
the edge of the Zagros orogeny. These spatial variations suggest that the observed bimodal
pattern of fast-axis orientations primarily results from lateral rather than vertical
heterogeneities in the anisotropic structure.



The fast axis orientations across the study region can be categorized into two main patterns.
First, in southern Iraq and the Persian Gulf (latitude < 32°N), the orientations are predominantly
NW-SE, sub-parallel to the Zagros orogeny and perpendicular to the absolute plate motion
(APM) direction within the no-net-rotation (NNR) reference frame (Kreemer et al., 2014; Fig.
1). This pattern is observed at stations BSR1, BSR2, NSR1, NSR2, NSR4, AMR1, AMR2,
SAM2, and KUT1 and extends into the Persian Gulf at station KHRK, marking the offshore
continuation of the Zagros foreland basin. Second, in northern Iraq (latitude > 33°N), the fast
axis orientations are primarily NE-SW, perpendicular to the Zagros orogeny and sub-parallel
to the APM vector, as observed at stations ANB1, ANB2, KIR1, SLY1, and DHK1. Station
KAR2, located near the boundary between these two regions, shows almost null SK(K)S
splitting, with 94 null measurements and 1 non-null measurements from a northeast back-
azimuth (Fig. 3 and Table 1). This suggests the presence of two dominant anisotropic features
with perpendicular orientations in the northern and southern sections of the study region. The
average splitting time is 0.82 s, which is lower than the values reported in surrounding regions,
including the Inner Arabian Platform, Anatolian Plate, and the Zagros collision zone, where
splitting times exceed 1 s (Paul et al., 2014; Qaysi et al., 2018; Kaviani et al., 2021), as shown
in Figure 5.

**4. Comparison with previous studies**
To investigate patterns of upper mantle seismic anisotropy, we combined our dataset with all
available SK[K]S measurements from the Middle East. This merging allows us to assess how
the results of this study align with or diverge from previous observations. Figure 6 presents
maps comparing our measurements with those from earlier studies in neighboring regions,
including the Iranian Plateau and eastern Anatolian Plateau (Kaviani et al., 2021), northwestern



Iran (Arvin et al., 2021), the northern Zagros (Sadeghi-Bagherabadi et al., 2018a, 2018b), and
the Inner Arabian Platform (Qaysi et al., 2018). In the left panels of Figure 6, individual
SK[K]S measurements are displayed, with red bars representing the new measurements and
blue bars indicating those from previous studies. Each measurement is projected onto its ray-
piercing point at depths of 100 km, 200 km, and 300 km, shown in panels a, c, and e,
respectively, to account for uncertainty in the depth of anisotropy.
Our results show a high degree of consistency with previous measurements, particularly in
southern Iraq and the Persian Gulf, where NW-SE-oriented fast axes are observed. Similarly,
in northern Iraq, the NE-SW-oriented fast axes align well with prior measurements in northern
Iraq and eastern Turkey (Kaviani et al., 2021). This agreement affirms that our findings extend
the observation of upper mantle anisotropy across the Zagros collision zone and enhance
understanding of mantle flow and deformation beneath this tectonically active region.
To further visualize the azimuthal anisotropy patterns, we calculated vector averages of the
individual splitting parameters projected at respective depths and within the Fresnel zone of
SK[K]S waves, utilizing sensitivity kernels as calculated by Monteiller and Chevrot (2011).
The interpolated anisotropy fields, shown in the right panels of Figure 6, demonstrate that the
anisotropic patterns remain largely consistent across different depth levels. This consistency
suggests a relatively simple and coherent anisotropic structure beneath the study region.

**5. Discussion**
**5.1 A uniform asthenospheric flow in the northern Middle East**
The anisotropy measurements in Figure 4 consistently show a fast axis orientation toward the
NE-SW in northern Iraq which is consistent with the NE-SW orientations in the Anatolian





Plateau and northwestern Iran (Fig. 6). This alignment closely correlates with the azimuth of
the APM direction in the NNR frame (Kreemer et al., 2014). The strong correlation across such
a vast region suggests the dominance of a large-scale viscous flow in the asthenosphere as the
prevailing mechanism beneath the northern Middle East (Sandvol et al., 2003; Paul et al.,

172    2014).

The dominance of asthenospheric flow is consistent with the presence of a relatively thin
lithosphere beneath the northern Middle East (Priestley and McKenzie, 2013), as illustrated in
Figure 7d. This correlation is significant because it indicates that the lithosphere has a minor
influence on the accumulated anisotropy responsible for the observed SK[K]S splitting. Further
evidence supporting this interpretation comes from S-receiver function analyses in
northwestern Iran and eastern Turkey, which reveal a thin lithosphere (ranging from 80 to 100
km in thickness) beneath eastern Anatolia, the Bitlis suture zone (e.g., Kind et al., 2015),
northwestern Iran (e.g., Taghizadeh-Farahmand et al., 2010), and northeastern Iran (e.g.,
Taghizadeh-Farahmand et al., 2013). Notably, the lithospheric mantle is entirely absent beneath
portions of the Anatolian Plate (Gök et al., 2007).
Furthermore, the alignment of the APM with the orientation of SK[K]S measurements (i.e.,
orientation of asthenospheric flow) suggests strong coupling between the thin lithosphere and
the underlying mantle in the northern Middle East. This observation reinforces the conclusion
that asthenospheric flow is the primary driver of plate tectonics in this region.
A comparison of the azimuthal anisotropy of Pn waves (Lü et al., 2017) with the fast-axis
orientations of SK[K]S waves projected to a depth of 75 km reveals different anisotropic
patterns across northern Iraq (Fig. 7c). This discrepancy suggests the presence of vertical
anisotropic layering, where distinct sources of anisotropy influence Pn and SK[K]S waves. We
propose that anisotropy within the lithospheric mantle has a limited influence on SK[K]S



splitting due to the thin lithospheric mantle in this region, while SK[K]S splitting primarily
reflects sub-lithospheric anisotropy associated with asthenospheric flow.
It is noteworthy that the tectonics of northern Iraq and eastern Turkey are dominated by the
active convergence between the Arabian Plate and Eurasia, accommodated by the westward
escape of Anatolia (Dewey et al., 1986; McClusky et al., 2000). Despite the complexity of this
tectonic junction, our azimuthal anisotropic patterns, as illustrated in Figures 4, 6, and 7, remain
relatively unaffected. This suggests that post-collisional tectonics in the region did not produce
a coherent or simple anisotropic texture in the lithosphere.

**5.2 Patterns of anisotropy beneath the Zagros Foredeep**
In southern Iraq and the Persian Gulf, the observed anisotropic fast-axis directions exhibit a
NW-SE orientation, perpendicular to the absolute plate motion and subparallel to the trend of
the Zagros orogeny. Observing these anisotropy orientations just behind a lithospheric root
beneath the Zagros orogeny (Fig. 7d) supports the concept of circular mantle flow around the
Zagros keel, as proposed by Kaviani et al. (2021). Their analysis of SK[K]S measurements in
the Iranian Plateau, covering the northern and eastern flanks of the Zagros orogeny, revealed
anisotropic axes encircling the Zagros lithospheric keel. This behavior, in which the lithosphere
acts as a barrier to horizontal asthenospheric flow when it is considerably thicker than the
surrounding regions, has been observed beneath the South American cratonic keel (Miller and
Becker, 2012) and eastern North America (Fouch et al., 2000).
At first glance, new data from the Zagros foreland in southern Iraq and the Persian Gulf appear
consistent with this model. However, a comparison of SK[K]S and Pn anisotropy in the same
region (Fig. 7c) reveals parallel orientations, indicating that at least part of the observed



anisotropy originates in the lithospheric mantle. Additionally, the splitting time beneath the
Mesopotamian Plain is significantly smaller than in surrounding regions, such as eastern
Anatolia and the Inner Arabian Platform (Fig. 5), where anisotropy is primarily driven by
asthenospheric flow (Hansen et al., 2006; Paul et al., 2014; Qaysi et al., 2018; Kaviani et al.,
2021). This suggests that while asthenospheric flow beneath the Mesopotamian Plain may still
generate anisotropy, its influence is significantly weaker than in nearby regions. This reduction
is likely due to the presence of a more complex asthenospheric flow near the lithospheric keel
beneath the Zagros orogeny (Fig. 7d). The lithospheric root causes substantial variations in
lithospheric thickness in adjacent areas, leading the asthenospheric flow dipping and being
deflected beneath the Mesopotamian Plain and the Zagros orogeny, as suggested by Sadeghi-
Bagherabadi et al. (2018b).
The dipping of asthenospheric flow can reduce the generation of azimuthal anisotropy, thereby
minimizing its contribution to the observed SK[K]S splitting in this region. In this scenario,
the small splitting times (Fig. 5) are partly attributed to the presence of a ~150 km thick
lithosphere beneath the region (Fig. 7d), with asthenospheric flow playing a limited role. An
alternative explanation is that the fast axis orientation from SK[K]S waves is parallel to the
slow axis within the lithospheric mantle, resulting in reduced splitting times recorded by
SK[K]S phases. Although our measurements do not reveal a clear pattern of asthenospheric
flow behind the lithospheric root beneath the Mesopotamian Plain, the absence of strong
asthenospheric flow in the observed splitting time offers insights into the regional flow
dynamics. Rather than rotating around the lithospheric root, asthenospheric flow may instead
migrate beneath it at greater depths. While this interpretation is not definitive, the lack of
significant splitting immediately behind the lithospheric root represents a novel observation
that challenges the concept of circular asthenospheric flow, suggesting a more complex flow
pattern.




### 5.3 Origin of Anisotropy in the Zagros Foreland Lithospheric Mantle

The NW-SE fast-axis orientation within the lithospheric mantle beneath the Mesopotamian
Plain and Persian Gulf raises questions about its origin. One possibility for the existence of
such anisotropy is that it results from pure shear deformation within the lithosphere due to the
continental collision in the Zagros orogeny. However, this scenario is unlikely because the
Arabian lithosphere is cold and strong, with temperatures below 900°C beneath the Moho
boundary (Priestley et al., 2012). Such low temperatures inhibit olivine mobility, preventing
alignment with the maximum strain direction (Nicolas and Christensen, 1987). An alternative
explanation is that the lithospheric mantle anisotropy reflects a "frozen" signature from earlier
tectonic events. Several significant tectonic episodes have shaped the region, including the
Precambrian Amar Collision (~640–620 Ma; Al-Husseini, 2000), the NAJD Rift System
(~570–530 Ma; Husseini, 1988, 1989; Husseini and Husseini, 1990), the Neotethys rifting
during the Triassic and Late Jurassic (e.g., Fadhel and Al-Rahim, 2019), and the ongoing
Zagros continental collision (~35 Ma to present; Jackson and McKenzie, 1984; Alavi, 2004).
The consistency of the NW-SE fast-axis orientation with the Zagros orogeny, its alignment
with the suture boundary between the Arabian and Eurasian plates, and its confinement to the
Zagros foreland depression (Fig. 4) suggest a connection to Neotethys rifting. Rift systems
elevate lithospheric temperatures, generating rift-parallel anisotropy aligned with the rift axis.
The Mesozoic rift axis between the Arabian and Eurasian plates was along the suture boundary
between the two plates. However, if diffuse rifting occurred between them, remnants of this
rift may also be preserved in structures parallel to the main axis.
Evidence for diffuse rifting in the region comes from previous geological and seismological
studies (Abdulnaby et al., 2020). Abdulnaby et al. (2020) analyzed the P receiver functions



beneath stations used in this study and found a crustal root beneath the southeastern
Mesopotamian Plain as thick as those in the Zagros collision zone. They proposed that the lack
of isostatic balance between the large crustal thickness and low topography in the region results
from successive rifting events. These events caused vertical loading from the accumulation of
thick sedimentary deposits in the basin, reaching thicknesses of up to 14 km over fault-bounded
depocenters, leading to the formation of crustal roots (Abdulnaby et al., 2020). The Abu Jir-
Euphrates fault is clearly observed as a basement fault step in seismic lines (Mohammed, 2006)
and bounds the southwest margin of the rift system (e.g., Fadhel and Al-Rahim, 2019). This
fault system, inherited from Triassic passive-margin extension, was reactivated during Middle
to Late Jurassic rifting, forming graben-horst structures in the Mesopotamian Plain (Numan,
1997, 2000). Continuous subsidence along these faults since the Late Jurassic allowed thick
sedimentary sequences to accumulate in this tectonic depression (Jassim and Göff, 2006).
In summary, our findings suggest that the uniform NW-SE fast-axis anisotropy within the
lithosphere of the Mesopotamian Plain likely originates from successive Mesozoic rifting of
the northeastern Arabian platform. This lithosphere has remained largely intact despite the
relatively young Zagros collision. The southwestward migration of the Zagros deformation
front (ZFF) overprinted the eastern boundary of the graben beneath the Mesopotamian Plain
through thrust faulting and folding with opposing dips in the sedimentary cover. Abdulnaby et
al. (2016a, 2016b) and Darweesh et al. (2017) proposed a southwestward dip of 60° for the
eastern margin of the basin beneath the ZFF. Collectively, these findings confirm that the
Mesopotamian Plain has remained largely unaffected by deformation from the Zagros collision
zone. Consequently, older tectonic events, such as Mesozoic rifting, are the most likely source
of the preserved lithospheric mantle anisotropy. This study provides the first evidence of such
rifting effects recorded in the lithospheric mantle.





**6. Conclusions**

This study presents SK[K]S splitting measurements from 18 newly deployed seismic stations in the foreland of the Zagros collision zone, filling a gap in the anisotropy map of the Middle East. Our dataset of 155 non-null measurements reveals two distinct fast-axis orientations: a NE-SW trend in northern Iraq and a NW-SE trend in the Mesopotamian Plain and Persian Gulf.

By integrating our results with recent anisotropy data from the Iranian-Anatolian Plateau, we identify a consistent NE-SW fast-axis orientation across the northern Middle East, including northern Iraq. This orientation closely aligns with the azimuth of absolute plate motion in the no-net-rotation reference frame, indicating that large-scale asthenospheric flow governs the observed anisotropy. The agreement between the absolute plate motion and asthenospheric flow suggests strong lithosphere-asthenosphere coupling, supporting the interpretation that asthenospheric flow is a key driver of plate dynamics in this region.

In contrast, the NW-SE anisotropy in the Mesopotamian Plain and Persian Gulf is associated with smaller splitting times, paralleling previously reported Pn anisotropy and suggesting a contribution from the lithospheric mantle. This finding challenges prior models of asthenospheric flow encircling the Zagros lithospheric keel, which implied weak lithosphere-asthenosphere coupling. Instead, the weak anisotropy beneath the Mesopotamian Plain may reflect steeply dipping asthenospheric flow beneath a laterally variable lithospheric thickness, resulting in reduced azimuthal anisotropy at the surface.

The NW-SE-orientated lithospheric anisotropy aligns with the suture boundary between the Arabian Plate and Eurasia and may represent a relic of diffuse Mesozoic rifting responsible for the formation of the Mesopotamian depression. These findings underscore the dual influence of asthenospheric flow and lithospheric deformation on seismic anisotropy in the Zagros



collision zone, offering new insights into mantle dynamics and lithosphere-asthenosphere
interactions in continental collision settings.
**Acknowledgements**
We are grateful to the Lawrence Livermore National Laboratory (LLNL) for supporting the
installation of broadband seismic stations in Iraq. We also like to thank the University of
Arkansas, Little Rock, for partially supporting this research.
**Data availability**
Continuous data from 17 Iraqi stations used in this study are available through the Incorporated
Research Institutions for Seismology (IRIS).
**Author contribution**
KM analyzed the data, prepared figures, interpreted the results, and wrote the initial draft of
the manuscript. AK developed the code, supervised the data analysis, prepared figures, and
revised the manuscript. WA conducted the field survey, collected raw data, provided the data,
and revised the manuscript. HM secured funding for data collection in Iraq and revised the
manuscript. HA secured funding for data collection in Iraq and revised the manuscript.
**Competing interests**
None of the authors has any competing interests.

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

**Figure 1**

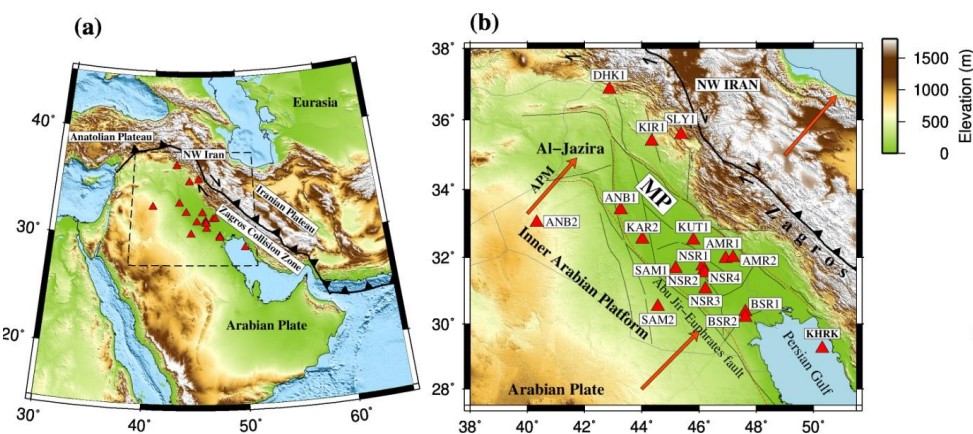



**Figure 1:** (a) Topographic Map of the Middle East. The triangles indicate the locations of 18
seismic stations within the study area. The solid lines mark the Bitlis-Zagros suture boundary.
The dashed-line rectangle outlines the boundaries of the map displayed in panel (b). (b)
Topographic Map of Mesopotamian Foredeep, situated in the foreland of the Zagros collision
zone. The red dashed line indicates the tectonic division of Iraq as proposed by Fouad (2010a,
2010b) and Sissakian et al. (2017), separating the Inner Arabian Platform from the Outer
Arabian Platform, which includes the Mesopotamian Foredeep. Al-Jazira, and Zagros collision
zone. Arrows represent the absolute plate motion (APM) vectors from Kreemer et al. (2014).
Thin black lines mark the location of basement faults within the Zagros Foreland Basin.
Topographic and bathymetric data were obtained from the ETOPO1 global relief model
(NOAA NCEI, 2022).





**Figure 2**

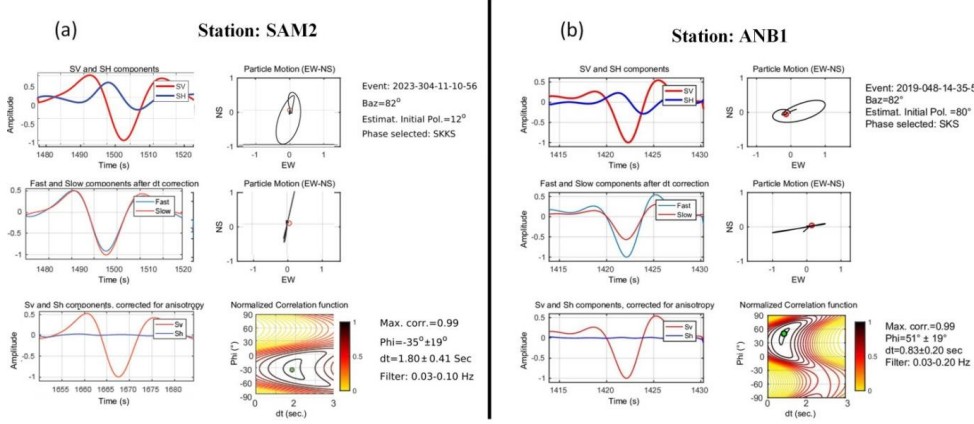


**Figure 2:** Examples of shear-wave splitting measurements using the rotation-correlation
method. The locations of the two stations can be inferred from Figure 1b. (a) Single-event
measurement at station SAM2. Data have been filtered to retain periods between 10 and 30 s.
Information about the event is provided in the top right corner. The left panels, from top to
bottom, display the original radial (red) and transverse (blue) seismograms, corrected fast
(blue) and slow (red) components, and corrected radial (red) and transverse (blue) components,
respectively. The right panels, from top to bottom, show the initial particle motion, the
corrected particle motion, and the contour plot of the normalized correlation function with the
optimal splitting parameter indicated by a green circle. The obtained splitting parameters are
written in the bottom right corner. (b) Similar to (a) but for station ANB1.



**Figure 3**
**Figure 3:** Rose plot of splitting measurements for stations used in this study. For each station,
non-null measurements are shown on the left-hand side plot as red bars oriented in the fast
direction with length proportional to the lag time. The initial polarization directions of null



measurements are shown as blue bars on the right-hand side plot. The locations of all stations
are shown in Figure 1.



**Figure 4**

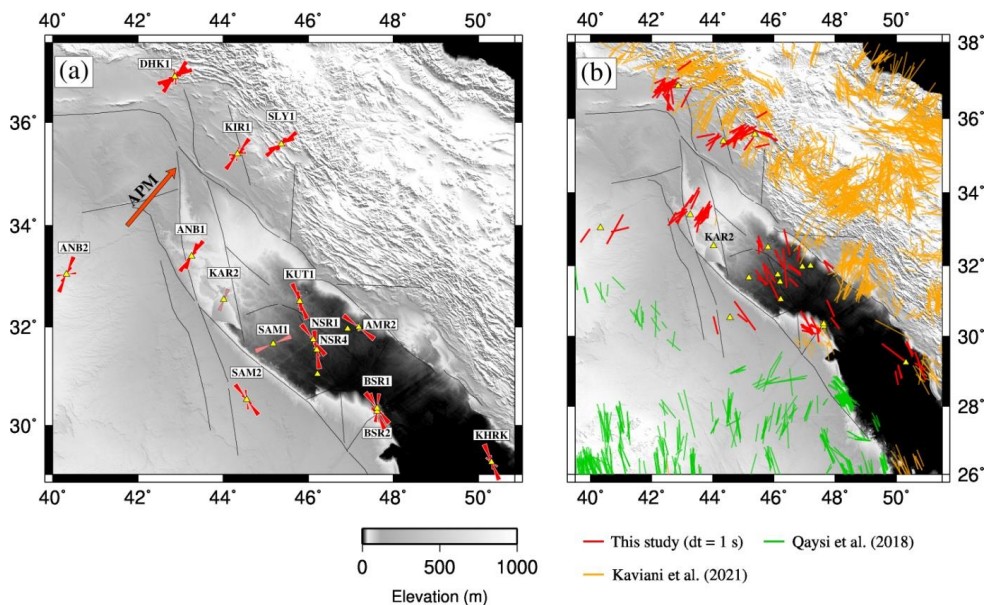


**Figure 4:** (a) Rose plot for non-null splitting measurements at seismic station locations
(triangles). Measurements for stations SAM2 and KAR2 are shown in pink, as each has only
one non-null observation. Arrow represent absolute plate motion (APM) vector from Kreemer
et al. (2014). (b) Individual fast-axis orientations from this study (red bars) and previous studies
by Qaysi et al. (2018) and Kaviani et al. (2021), projected onto the ray-piercing points at a
depth of 200 km. Topographic and bathymetric data were obtained from the ETOPO1 global
relief model (NOAA NCEI, 2022).





**Figure 5**

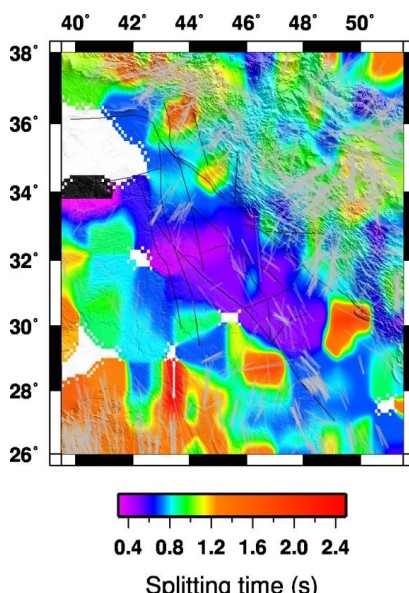


**Figure 5:** Distribution of spatially averaged splitting times within the Zagros Foreland and
surrounding regions. Thin gray bars represent the anisotropy fast axis orientations, with their
lengths scaled to match the splitting times.



**Figure 6**

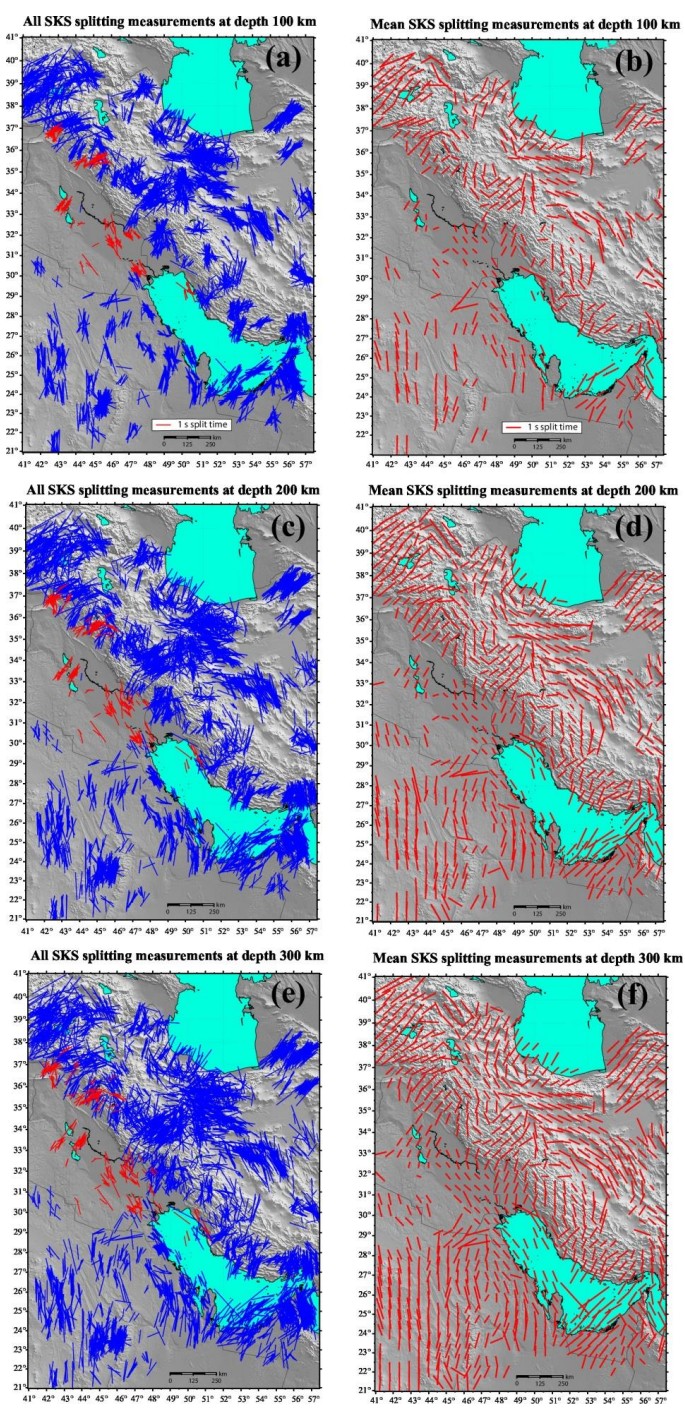




**Figure 6:** Anisotropic fast axis orientations from the current study (depicted by red bars in the
left panels) combined with prior measurements (represented by blue bars) from Kaviani et al.
(2021), Arvin et al. (2021), Sadeghi-Bagherabadi et al. (2018a and 2018b), and Qaysi et al.
(2018). The left panels illustrate the fast axis orientations projected onto the ray-piercing point
at depths of (a) 100 km, (c) 200 km, and (e) 300 km. The right panels display interpolated
anisotropy fields at depths of (b) 100 km, (d) 200 km, and (f) 300 km. Elevation data were
derived from ETOPO1 (NOAA NCEI, 2022).
**Figure 7**

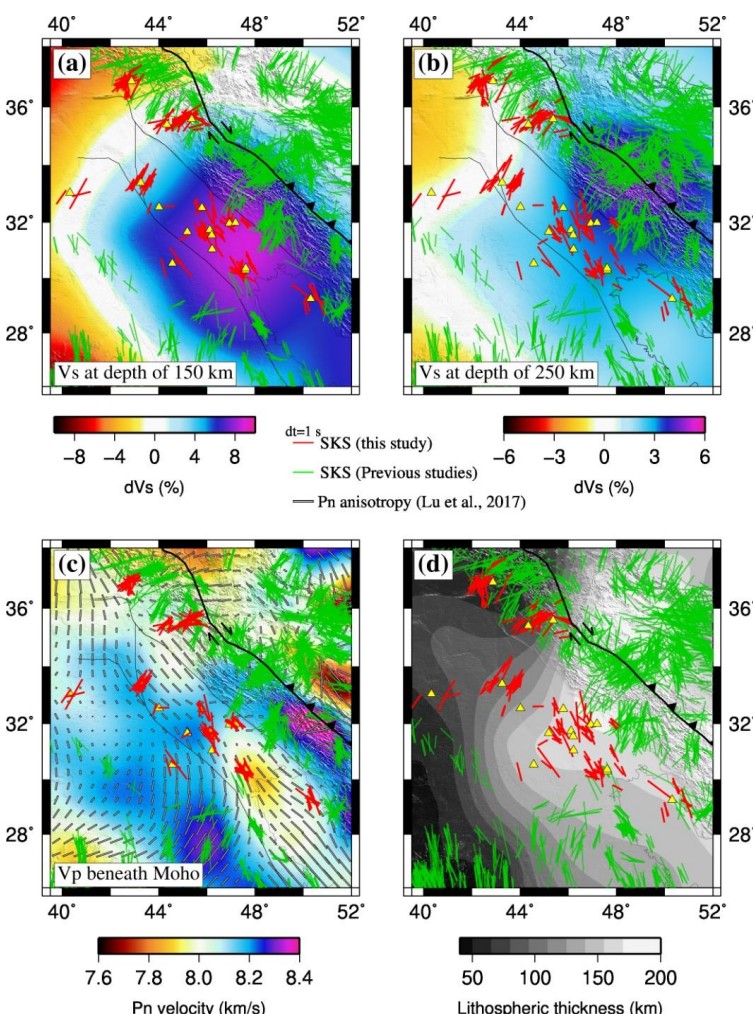


**Figure 7:** (a) Shear-wave velocity (Vs) map at a depth of 150 km from regional full-waveform

tomography by Celli et al. (2020). Colored bars represent individual fast-axis orientations from

this study and previous studies, projected onto ray-piercing points at a depth of 150 km. Thin

black lines mark the borders of Mesopotamian Foredeep and Al-Jazira (Sissakian et al., 2017).

(b) Same as (a), but at a depth of 250 km. (c) Pn velocity map with fast-axis anisotropy

orientations (gray bars) from Lu et al. (2017), overlaid with individual fast-axis orientation

measurements projected to a depth of 75 km. (d) Lithospheric thickness map from Priestley



and McKenzie (2013), with colored bars representing the individual fast-axis orientations
projected onto ray-piercing points at a depth of 250 km. Elevation data were derived from
ETOPO1 (NOAA NCEI, 2022).


**Table 1.** Summary of the used stations in this study and their splitting parameters. The table
shows the station location, the circular mean of the fast axis orientation ($\overline{\varphi}$), the mean splitting
time ($\overline{\delta t}$), the number of splitting measurements (SM), and the number of null measurements
(NM). Stations with bimodal fast axis orientations are marked with an asterisk (*).

| Station | Latitude | Longitude | Begin date (YYYY/MM) | End date (YYYY/MM) | $\overline{\varphi}$ (o) | $\overline{\delta t}$(s) | SM | NM |
|---------|----------|-----------|----------------------|--------------------|------|------|----|----|
| AMR1 | 31.9590 | 46.9286 | 2015/03 | 2015/10 | - | - | 0 | 8 |
| AMR2 | 31.9899 | 47.1902 | 2015/11 | 2022/08 | -42º | 0.65 | 8 | 52 |
| ANB1 | 33.401 | 43.2576 | 2018/10 | present | 32º | 0.76 | 31 | 44 |
| ANB2 | 33.0375 | 40.320 | 2023/06 | present | 42º | 1.15 | 3 | 5 |
| BSR1* | 30.3581 | 47.6153 | 2014/08 | 2015/08 | -19º | 0.61 | 2 | 8 |
| BSR2 | 30.2927 | 47.6191 | 2015/09 | present | -43º | 0.70 | 16 | 93 |
| DHK1* | 36.8606 | 42.8665 | 2014/01 | present | 38º | 0.81 | 32 | 74 |
| KAR2 | 32.5398 | 44.0224 | 2017/01 | 2023/02 | 17º | 0.52 | 1 | 94 |
| KIR1 | 35.388 | 44.3419 | 2018/09 | 2021/08 | 44º | 0.74 | 7 | 17 |
| KHRK | 29.2543 | 50.3133 | 2021/04 | present | -29º | 0.86 | 9 | 7 |
| KUT1 | 32.509 | 45.797 | 2021/11 | 2023/02 | -19º | 0.87 | 6 | 32 |
| NSR1 | 31.7416 | 46.1151 | 2014/08 | 2017/09 | -35º | 0.69 | 6 | 40 |
| NSR2 | 31.5550 | 46.1374 | 2014/07 | 2014/09 | - | - | 0 | 3 |
| NSR3 | 31.0514 | 46.2199 | 2014/07 | 2014/11 | - | - | 0 | 2 |
| NSR4 | 31.540 | 46.202 | 2017/10 | present | -16º | 1.02 | 5 | 68 |
| SAM1 | 31.661 | 45.183 | 2020/12 | 2021/11 | 60º | 0.51 | 1 | 6 |
| SAM2 | 30.5295 | 44.5587 | 2023/03 | present | -33º | 1.12 | 3 | 14 |
| SLY1 | 35.5784 | 45.3667 | 2015/09 | present | 38º | 0.98 | 25 | 63 |

