# Peer review of "Seismic anisotropy under Zagros foreland from SKS splitting observations"

_EGUsphere, 2025_

## Author Comment (AC1)

**Response to Reviewers' comments on the manuscript egusphere-2025-1231**

**Seismic anisotropy under Zagros foreland from SKS splitting observations**

*by Khalil Motaghi, Ayoub Kaviani, Wathiq Abdulnaby, Hanan Mahdi, Haydar Al-Shukri*

We thank both reviewers for their constructive and insightful comments, which significantly improved the clarity, scientific rationale, and interpretation of the results presented in our manuscript. Below, we address the reviewers' comments point by point.

**Please note:**

- Reviewers' comments are in black, replies in blue. Text added to (or modified within) the manuscript is repeated here in italics and quotes.
- In the annotated manuscript, added or modified text is marked in red.
- Unless otherwise stated, line number references below are in relation to the revised manuscript.
* * *
**Reviewer 1 – Response to Comments**

**1. Abstract – General statement about refining mantle dynamics is too generic. Request to clarify specific tectonic-geodynamic findings.**

**Reviewer comment:**

These findings refine our understanding of mantle dynamics and lithosphere-asthenosphere interactions in the Zagros collision zone.

This is a rather generic statement... Authors must emphasize their findings on what new tectonic-geodynamic concluding remarks this study brings...

**Response:**

We thank the reviewer for this helpful suggestion. We agree that the concluding sentence of the abstract was too general. We have revised it to more clearly reflect the specific findings of this study, emphasizing the recognition of lithospheric anisotropy linked to Mesozoic rifting in the Mesopotamian Plain and the limited role of asthenospheric flow in this region. The revised sentence now reads:

*"These findings demonstrate a dual origin of seismic anisotropy in the Zagros foreland, where lithospheric fabric related to Mesozoic rifting dominates in the south, while*

*asthenospheric flow governs the anisotropy in northern Iraq, refining models of mantle dynamics and lithosphere-asthenosphere coupling in continental collision zones."*

**2. Introduction** – Lacking scientific rationale and motivation.

**Response:**

We appreciate this important observation. In the revised manuscript, we have thoroughly revised the Introduction to better articulate the scientific rationale and significance of the study. We highlight the lack of seismic anisotropy constraints in the Zagros foreland and underscore the critical role of mantle fabric in understanding the geodynamic evolution of the Arabia-Eurasia collision zone. Specifically, we have added the following objectives at the end of the Introduction (Lines 79-83):

"

- *To determine whether seismic anisotropy in the Zagros foreland originates in the lithosphere or asthenosphere;*
- *To assess the possible deflection or suppression of mantle flow by the lithospheric keel beneath the Zagros;*
- *To investigate whether the NW–SE anisotropy observed in the Mesopotamian Plain reflects fossil fabric from Mesozoic rifting.*

"

3. "Frozen anisotropy" hypothesis lacks modeling-based support.

**Response:**

We acknowledge the reviewer's suggestion that the interpretation of frozen lithosphere anisotropy would benefit from additional modeling to constrain symmetry types and orientations. However, our main argument is based on consistent SKS fast-axis orientations, comparison with regional tectonics, and alignment with rift structures. We agree this remains an interpretive component and have adjusted the wording in the revised Discussion accordingly.

4. Request for number of earthquakes used (not just records).

**Response:**

We thank the reviewer for pointing this out. We have now added the total number of teleseismic events (342 out of 1240) used in the analysis to the „Data and method" section (line 94).

5. Question about the 50% energy reduction threshold.

**Response:**

We agree that the 50% threshold is a moderately permissive criterion, chosen to ensure a balance between data quantity and measurement reliability, especially in regions with limited seismic coverage. We tested more stringent thresholds (70–80%) but found a significant drop in usable measurements, particularly from stations in noisy or remote areas. We now clarify this point in the relevant part (lines 111-114) in the "Data and method" section:

*"Although stricter thresholds (e.g., 70–80%) would enhance robustness, preliminary tests showed they significantly reduced the number of non-null results, especially at stations with lower signal quality. The 50% threshold was thus adopted as a compromise to balance spatial coverage and measurement reliability."*

6. Line 102–104: Use "SH" instead of "Sh".

**Response:**

We have corrected all instances of "Sh" to "SH" in the manuscript.

7. Recommend using multiple SKS splitting methods (e.g., minimum energy).

**Response:**

We appreciate this valuable recommendation. While our main analysis used the rotation-correlation method, we have now implemented a comparison on a subset of events using the minimum energy method. The results were presented in new Table S1 (presented in the supplemental material) and they show high consistency in splitting parameters for high-quality events. Due to computational constraints and data volume, we did not re-process all data using both methods, but we now acknowledge this in the "Data and method" and have added a paragraph (lines 120-124):

*"To evaluate the robustness of our results, we reanalyzed a representative subset of events using the minimum energy method. The results (Table S1, in supplementary material) showed*

*good agreement with those obtained from the rotation-correlation technique. Future work incorporating full-matrix comparisons of all events using multiple methods is warranted."*

8. How are uncertainties estimated?

**Response:**

Thank you for raising this point. We now describe our approach for estimating uncertainties in a separate paragraph in the „Data and method" section. We added (lines 116-119):

*"Uncertainties in fast-axis orientation and splitting time were estimated using the contour method, based on 95% confidence regions from the normalized correlation surface. The standard deviation of splitting parameters within this region was used as the uncertainty measure for each event."*

9. Line 119–120 and 162: Interpretative language in Results section.

**Response:**

We agree with the reviewer that interpretative statements should be confined to the Discussion section. We have revised these sentences in the Results section to present only objective observations. The relevant interpretative remarks were moved to the Discussion.

10. Null measurements underutilized; recommend providing additional information and discussion.

**Response:**

We appreciate the reviewer's comment regarding the underuse of null SKS splitting measurements in our discussion.

In the revised Result section, we analyzed the null data set and compared their azimuthal distribution with the non-null measurements using the new rose diagrams presented as new Figure 5 (presented below too as Figure R1). The corresponding explanation in the revised manuscript appears in lines 157–173:

"*An important question is why a large number of null measurements (630 out of 785) are observed. Figure 5 addresses this by showing polar histograms of back-azimuths for null and non-null measurements across the dataset. Two dominant directions are associated with nulls: east and southwest (≈210°). The southwestern azimuth coincides with the fast axis orientation in northern Iraq and the slow axis in southern Iraq, suggesting that SK[K]S*

*phases from this direction are polarized along the symmetry axes of anisotropy, resulting in null splitting. However, the most prominent direction for nulls is from the east. Station-specific back-azimuth rose diagrams (Fig. 3, right panels) show that stations AMR2, BSR2, NSR1, NSR4, KUT1, KAR2, and KHRK, all located in the Mesopotamian Plain and Persian Gulf, exhibit a high concentration of nulls from the east. In contrast, stations in northern Iraq (e.g., ANB1, ANB2, KIR1, and SLY1) do not show a similar eastern null pattern. This systematic azimuthal dependence likely results from both the complex lithospheric structure east of the Mesopotamian Plain, where it interacts with the Zagros, and the global distribution of teleseismic sources. A significant number of SK[K]S phases arrive from the east, where active subduction zones along the western Pacific margin fall within the optimal epicentral distance range for splitting analysis. Non-null measurements, by contrast, predominantly arrive from other directions, especially from the west, as seen in Figure 3."*

[Figure]

**Figure R1.** New Figure 5 for the revised manuscript. It presents *rose plots of back-azimuths for (a) null and (b) non-null measurements, binned in 10° sectors. Red lines indicate the fast axis orientations of anisotropy in northern and southern Iraq. N denotes the number of measurements used to plot the rose diagrams.*

In the revised Discussion section, we now emphasize the weak splitting times observed beneath the Mesopotamian Plain, and we clarify that most of the null measurements are concentrated at one station in this region.

We acknowledge the interpretational complexity of nulls, which can reflect vertical symmetry axes, isotropy, or the cancellation effects from multi-layered anisotropy. However, given the limited number and spatial clustering of null observations in our dataset, we believe that an

extended interpretation would be speculative. We therefore mention them where relevant but refrain from detailed interpretation in the absence of broader supporting evidence.

The text added to the end of Discussion section (lines 312-318) was presented below for reviewer reference:

*"The high occurrence of null measurements from the east may indicate a complex interaction between the lithosphere and asthenosphere at the boundary between Mesopotamia and the Zagros. A plausible explanation for these nulls is a two-layer anisotropic structure with orthogonal fast orientations, where the opposing effects of each layer interfere destructively. Such a configuration could result in the observed nulls for eastward-arriving waves across the Mesopotamian plain."*

11. Table 1 – Errors on splitting parameters missing.

**Response:**

We have now included the standard deviations ($1\sigma$) of the fast-axis orientations and delay times in Table 1, calculated from the available measurements at each station. These values are reported after the ± symbols.

12. Fig. 5 – Clarify interpolation method and address artifacts.

**Response:**

We have updated the map shown in Figure 5 (Figure 6 in the revised manuscript). We also modified the text accordingly and explained how the map was generated (lines 174-181). We now emphasize that the split time is generally lower in the Zagros foreland area compared to the surrounding regions. In addition, we repeat in the figure caption that the map is generated by resampling the split times at regularly spaced 1° grid points, averaging over the Fresnel zone around each point, and linearly interpolating between the grid points.

[Figure]

**Figure R2.** Revised version of Figure 6 from the manuscript. It shows *spatial distribution of SKS splitting times across the Zagros foredeep and surrounding regions, based on data from this study and previous works. The map is generated by resampling the splitting times at regularly spaced 1° grid points, averaging over the Fresnel zone around each point, and linearly interpolating between the grid points.*

13. Fig. 7d – Correlation between null measurements and LAB depth.

**Response:**

In the revised Discussion section (Subsection 5.2), we expand the discussion on the correlation between the anisotropy pattern and lithospheric thickness (Fig. 8d). We observe a moderate spatial correlation between regions of thicker lithosphere (>150 km) and a higher

incidence of null measurements, particularly in the vicinity of the Mesopotamian Plain. This supports the interpretation that vertical anisotropy or weak asthenospheric flow beneath thick lithosphere may be responsible for the reduced splitting signals.
* * *
**Response to Reviewer 2**

**General Comments**

This manuscript presents new azimuthal anisotropy measurements... At least, that's what I think is the main aim of this work. Unfortunately, the scientific objectives are not clearly stated... The final discussion is confusing and convoluted...

**Response:**

We agree with the reviewer that the scientific objectives were not clearly articulated in the original version. In response, we have rewritten the **Introduction** section  (see also **Reviewer 1, Comment 2**).

We have also restructured the **Discussion section** to align with these scientific objectives, significantly improving its focus and readability.

**Specific Comments**

1. Abstract lacks rationale; Why is it important to have more measurements in the foreland?... The last sentence is vague.

**Response:**

This point was also raised by **Reviewer 1 (Comment 1)**. The Abstract has been revised to highlight the absence of previous SKS splitting observations in the Zagros foredeep and to emphasize its tectonic significance.

2. Introduction should begin with aims; add map of existing measurements

**Response:**

As noted above and in **Reviewer 1 (Comment 2)**, we now begin the Introduction with a concise statement of the motivation and objectives of our study.

3. Line 63 – you mention "the thick lithospheric root" but references are missing; is this a widely accepted observation?

**Response:**

We have added appropriate references here, including: Priestley et al. (2012).

4. Figure 1 – I cannot see the red dashed line. Scale of APM arrows missing.

**Response:**

We revised Figure 1 to increase the visibility of the red dashed line and added a scale bar for the APM vectors. The caption was updated accordingly. The revised figure is presented below.

[Figure]

**Figure R3.** Revised version of Figure 1 from the manuscript. In this figure, red dashed lines and a scale bar for the APM vector have been added.

5. Figure 3 – what are the grey sectors? Could you add the numbers of non-null and null measurements at each station on this figure?

**Response:**

The gray wedges in Figure 3 represent histograms of the non-null individual measurements, binned in 15° sectors. They were included to simplify the presentation by summarizing the observations in histogram form. Some of the shorter gray wedges may not be clearly visible as they are obscured by the overlying red bars. we added the following sentence to the figure caption to explain the gray sectors:

*"The gray wedges represent histograms of the individual measurements, binned in 15° sectors."*

We also added the number of null measurements (NM) and number of splitting measurements (SM) to revised version of Figure 3 as requested. The revised figure was presented below.

[Figure]

**Figure R4.** Revised version of Figure 3 from the manuscript.

6. Line 113 – Lack of commentary on null-only stations

**Response:**

This concern is also addressed in **Reviewer 1 (Comment 10)**. We now include a brief explanation in the Result section (lines 157-173) and discussion of null measurements at the end of Discussion section in lines 312-318.

7. Lines 134–137 – is the difference in average splitting time between 0.82s and >1s statistically significant? The map of Fig. 5 documents this difference, but explanations on how it is computed are missing.

**Response:**

As noted in **Reviewer 1 (Comment 12),** we have updated the map shown in Figure 6 (or Fig. R2). We also modified the text accordingly (lines 174-181). We now emphasize that the split time is generally lower in the Zagros foreland area compared to the surrounding regions. We explain in the result section and also figure caption that the map is generated by resampling the split times at regularly spaced 1° grid points, averaging over the Fresnel zone around each point, and linearly interpolating between the grid points. Here is our explanation in lines 174-181:

"*Figure 6 maps the lateral variation of SK[K]S splitting times across the study area and surrounding region, incorporating results from this study with those from previous studies. The map is generated by resampling the splitting times at regularly spaced 1° grid points and averaging them over the Fresnel zone around each point. The final map is then produced by linearly interpolating between the grid points. The results show that splitting times beneath the Zagros foreland are generally much smaller than those reported for the surrounding regions, including the Inner Arabian Platform, the eastern Anatolian Plate, and the Zagros collision zone (Paul et al., 2014; Qaysi et al., 2018; Kaviani et al., 2021)."*

8. Line 162 – Clarify "simple anisotropic structure"

**Response:**

This comment overlaps with **Reviewer 1 (Comment 9)**. This part of Discussion was thoroughly modified in the revised version.

9. Line 167 – Only part of Anatolia shown in Fig. 6: you only show Eastern Anatolia and not the whole of it.

**Response:**

Figure 7 already covers a broad region surrounding our study area. While it does not include the entirety of the Anatolian Plate, we believe that referencing Paul et al. (2014) sufficiently supports our discussion of NE–SW-oriented SKS anisotropy in Anatolia. Including the full extent of their results in our figure is not essential for conveying our main arguments, as their findings are well established and appropriately cited where relevant.

10. Figure 7 – should show the entire regional area and not only the Zagros foreland.

**Response:**

As also noted by **Reviewer 1 (Comment 13)**, **Figure 7** has been updated to:

- Show a **wider regional extent**
- Include the **approximate outline of the Zagros lithospheric keel,** based on Priestley et al. (2012)

The updated figure was presented below. We also discuss the spatial relationship between anisotropy patterns and the keel more clearly in the revised Discussion (lines 265-273).

[Figure]

**Figure R5.** Revised version of Figure 8 from the manuscript.

11. Lines 169, 171, 175 – Clarify terms like "correlation," "Northern Middle East"

**Response:**

We substantially revised the text, resulting in significant changes. The discussion section 4.2

now opens with: 'In the northern Arabian Plate and much of the Eurasian Plate, the fast axes of SK[K]S splitting are consistently aligned with the APM direction of the Arabian Plate (Figs. 4 and 7).' Here, we intentionally use 'consistent alignment' rather than 'correlation.' Furthermore, instead of referring to the 'Northern Middle East,' we specify 'the northern Arabian Plate and much of the Eurasian Plate,' as shown in Figs. 4 and 7.

12. Line 184: The alignment with APM suggests a coupling between the surface (where the APM is measured) - not the lithosphere - and the asthenosphere.

**Response:**

The absolute plate motion at the surface is often assumed to represent the movement of the lithosphere as a rigid plate.

13. Figure 7c – Comparison with Pn anisotropy: Fig. 7c does not reveal different anisotropic patterns but coherences and inconsistencies between Pn velocity anisotropy and SKS anisotropy. No scale of Pn anisotropy is shown in Fig. 7c. Is it significant?

**Response:**

We added a **scale bar** to Figure 8c and added "3%" above the gray line in the figure legend to make the scale clear. The Discussion now emphasize on the coherence and divergence between SKS and Pn anisotropy, which highlights possible vertical complexity in anisotropy structure.

14. Line 192 – Pn comparison may be redundant. When you know that the lithosphere is thin, you don't need to compare to Pn to state that the lithosphere has a negligible influence on the total anisotropy. The asthenospheric origin of SKS anisotropy is now widely accepted, isn't it?

**Response:**

As noted by both reviewers, we have revised the paragraph to acknowledge that asthenospheric origin of SKS anisotropy is widely accepted, and that **Pn anisotropy is used here as a vertical comparison tool**, not as a primary diagnostic.

15. Line 196 – what do you mean by "complexity of this tectonic region"? The APM vectors rather document a simple pattern.

**Response:**

We modified the relevant paragraph with a more specific statement regarding **structural variation in the lithosphere and upper mantle** due to the ongoing collision.

16. Line 198 – "unaffected" by what?

**Response:**

As stated in response to the previous comment, we have thoroughly revised the relevant paragraph. In the revised version, we emphasize that despite the complex tectonic setting resulting from the convergence of the Arabian and Eurasian plates and the westward escape of Anatolia, the consistent anisotropy patterns in northern Iraq and eastern Turkey suggest that asthenospheric flow, rather than lithospheric deformation, is the primary source of seismic anisotropy.

17. Line 198 – what do you mean by "post-collisional tectonics"? The Zagros collision is still ongoing.

**Response:**

The entire subsection has been revised. In the updated version, we no longer use the term "post-collisional tectonics".

18. Line 204 – what do you mean by "behind a lithospheric root"? According to Fig. 7d, these measurements are located in the thick lithosphere region.

**Response:**

This part was completely modified in the revised manuscript.

19. Line 205–208 – the circular mantle flow is not visible in Fig. 7d, which is too zoomed. If the mantle flow turns around the keel, there should be no asthenospheric flow beneath the keel. Can you plot the extent of the keel in the Fig.? Fig. 7d shows abrupt changes in fast-velocity directions between the Mesopotamian Foredeep and the region of the Zagros suture. How do you explain them?

**Response:**

In the revised version, we discuss how the thick lithospheric "keel" beneath the Mesopotamian Plain disturbs large-scale asthenospheric flow: diverting it around the keel and inhibiting the development of a coherent anisotropic fabric beneath it.

20. Line 286 – "First evidence" overstates findings: "..the first evidence of such rifting effects...": rather a hypothesis than an evidence.

**Response:**

We agree with you and have removed this sentence, as it is a hypothesis rather than evidence.

---

## Author Response (AR2)

**Response to Reviewers' comments on the manuscript egusphere-2025-1231**

**Seismic anisotropy under Zagros foreland from SKS splitting observations**

*by Khalil Motaghi, Ayoub Kaviani, Wathiq Abdulnaby, Hanan Mahdi, Haydar Al-Shukri*

We are grateful for the attention of the associate editor to our manuscript and for second review by the reviewer 1 and their helpful comments.

**Comment by the reviewer:**

This is my second iteration in reviewing this work. Authors' answers and actions taken for my comments are usually satisfactory. However, my critic for %50 of threshold still need to convince the reader to make sure the reliability of the results. Thus I would like to see the all individual splitting measurements to have a feeling how elliptical particle motion is corrected back to linear particle motion after correcting for anisotropy during the splitting measurements.

So it is better for me to check all of the individual measurements to feel OK.

**Response:**

Following our second revision, we also consider that this part was not clearly explained in the previous version of the manuscript. Therefore, we have modified this part (lines 110–121 in clean version) based on the analysis of energy distribution of the T-component seismograms before and after correction for the effect of shear-wave splitting. The results of this analysis are shown as Figure S1 in supplementary information. These figures are also included here.

[Figure]

As shown in this Figure (panel b), reductions of T-component energy up to 95% are observed, but most measurements cluster between 50% and 60%. The T/R energy ratio is already low before correction (panel a, blue charts), so only moderate energy reduction is expected after

correcting for anisotropy. The post-correction T/R energy distribution  (panel a, orange charts) further confirms that low T-component amplitudes are left, indicating linear R-T particle motion after correction for anisotropy.

All individual measurements are provided in Table S2 of the supplementary information.